# Effects of Acute Visual Stimulation Exercise on Attention Processes: An ERP Study

**DOI:** 10.3390/ijerph18031107

**Published:** 2021-01-27

**Authors:** Shanshan Wu, Hongqing Ji, Junyeon Won, Xiaolong Liu, Jung-Jun Park

**Affiliations:** 1School of Physical Education & Health, Wenzhou University, Wenzhou 325035, China; wss1229@wzu.edu.cn (S.W.); jhq0825@wzu.edu.cn (H.J.); 2Division of Sport Science, Pusan National University, Busan 46241, Korea; 3Department of Kinesiology, University of Maryland, College Park, MD 20740, USA; won25@umd.edu; 4Institute of Brain and Psychological Sciences, Sichuan Normal University, Chengdu 610066, China; xiaolongliu@sicnu.edu.cn; 5The Clinical Hospital of Chengdu Brain Science Institute, MOE Key Laboratory for Neuroinformation, University of Electronic Science and Technology of China, Chengdu 611731, China

**Keywords:** acute visual stimulation exercise, P2, N2b, P3a, attention

## Abstract

Backgrounds: It remains to be determined whether visual stimuli during exercise differentially influence the attention process. The purpose of the present study was to examine if different color stimuli during aerobic exercise are associated with different attention processes. Methods: 22 college students completed a four 30-min running session during the presentation of different color stimuli (blue, green, red, and yellow) and without color stimulus on separate visits. The Kanizsa triangle task was administrated before and immediately after exercise to assess the attention process. Behavioral performance (accuracy and response time (RT)) and event-related potential (P2, N2b and P3a) were recorded during the test. Results: Valid/invalid cue RT during the Kaniza test performance was significantly faster following the presentation of color stimuli during treadmill exercise compared to the seated rest. During exercise, these changes were larger after green and yellow stimuli than red in invalid cue RT. P2, N2b and P3a amplitudes of green were significantly larger than the other colors for both valid and invalid cues. Red color showed the lowest P2 and P3a amplitudes for both valid and invalid cues among colors. Conclusion: The distinctive neurocognitive changes during aerobic exercise suggest different effects of color stimuli on visual search attention, attention capture, attentional orienting and processing speed. This study will be a first step to understand the optimal environmental setting during exercise for subsequent improvements in the attention process.

## 1. Introduction

Attention is the ability to choose and concentrate on relevant stimuli and to make consequent responses [1]. Therefore, attention is an essential function in our daily lives. A growing number of studies demonstrates that colors act as attentional stimuli either inducing distraction [2] or directing attention [3,4,5,6]. Colors appear to have different effects on the attention system through its differential emotional valence and attentional priority [7,8,9]. Clarke and Costall [10] reported green, blue and purple attenuate anxiety levels and physiological stress, while red, yellow, and orange elicit emotional stimulation. A past electroencephalography (EEG) work suggested that brain waves respond differently to color stimuli; green increases the alpha power, indicating greater attention; yellow increases theta power, suggesting greater happiness [11].

Beyond the observation of alpha and theta powers, the EEG activity during visual stimulus tasks serves as an indicator of specific cognitive operations. Event-related potential (ERP) is EEG signals assessing the electrical responses generated within the cortex during processing visual or cognitive events [12]. The P2 component is a positive deflection, prominent over the central electrode, with a typical peak latency of approximately 120–270 ms induced by visual stimuli. The P2 has been also implicated in visual search attention, language context information, and memory and repetition effects [13]. The N2b component is the most prominent over the central electrode, with a typical peak latency of approximately 200–380 ms. The N2b involves in orienting attention [14], attentional capture [15], and identification of stimuli [3]. During visual selective attention tasks, the N2b increases at the observation of target stimuli [3,15]. The P3a component is a positive scalp-recorded brain potential that has maximum amplitudes over frontal/central electrode sites, with a peak latency falling in the range of 250–280 ms [16]. The P3a reflects brain activity related to the engagement of attention (especially orienting and involuntary shifts to changes in the environment) and the processing of novelty [17]. 

It is well-documented that acute exercise engenders cognitive benefits during various cognitive tasks [18,19,20,21,22,23,24,25]. For example, a moderate-intensity aerobic exercise generally increased the allocation of attentional resources (increased P3b amplitude) [18,26] and increased N2 amplitude during the flanker task, suggesting enhanced inhibitory control [27]. In addition, selection and giving priority to incoming stimuli are important factors for the higher quality of exercise and sports performance [28]. According to Tsai and colleagues, activities with open-skill (e.g., racket sports, team sports) resulted in a larger P3 amplitude and shorter response time (RT) during spatial attention test than closed-skill (e.g., jogging, swimming, cycling), indicating that open skills are associated with enhanced spatial attention compared to closed skills [29]. 

In prior ERP works, the effects of exercise on attention have been predominantly studied by using the P300 component. Only a few studies have focused on other ERP components such as P2 and N2b components. Furthermore, given that external stimuli during exercise significantly affect visual–spatial attention [30], understanding the optimal environment to enhance spatial attention during exercise is also significant. Therefore, we investigated whether color stimuli during an acute bout of exercise influences the functioning of the attention processes using attention task performance and P2, N2b, and P3a responses. Based on prior findings that showed the effects of green color on increasing attention [11], we hypothesized that green will demonstrate greater improvement in spatial attention performance and EEG-related responses compared to other colors and no color.

## 2. Materials and Methods 

### 2.1. Participants

Twenty-two healthy, sedentary young adults (ages 21–27 years) participated in the present study. Subject inclusion criteria were: (1) age between 20 and 30 years, (2) right-handedness, and (3) normal or corrected-to-normal vision and without color-blindness. All participants were cognitively normal and did not take neurological medication or report neurological problems. Participants were instructed to refrain from physical activity on experimental visits. The participants were also instructed to maintain normal eating habits, to avoid eating for four hours, drinking alcohol for 12 h, and drinking caffeine for four hours prior to participation each day. Prior to the first experimental session, eligible participants provided informed consent approved by the Institutional Review Board of Pusan National University. This study was conducted in accordance with the Helsinki Declaration. Demographic information of all participants is provided in Table 1.

### 2.2. Experimental Procedure

This study consisted of six experimental interventions (e.g., seated-rest and five exercise conditions including no-color, red, blue, green, and yellow). We used a counterbalanced crossover design where participants were randomly assigned to perform treadmill running during exposure to either blue, green, red, or yellow stimulus. Participants also performed the seated-rest, treadmill running without color stimulus as a control condition. To minimize the possibility of learning and habituation effects, the tests were separated by intervals of greater than seven days. Before and after each exercise, attention processes were tested using the Kanizsa triangle task during which behavioral performance (accuracy and RT) and electrophysiological signals (P2, N2b, and P3a amplitudes) were collected. Each exercise consisted of 20 min of continuous treadmill running at 60–80% of maximum heart rate (HR). The exercise began with a 5-min warm-up and cool-down at 40% of maximum HR. After exercise, participants performed a 10-min seated rest before administering EEG measurement when the participant’s HR returned to pre-exercise levels. HR during exercise was monitored using HR monitors (polar RS400sd, Madison Height, Michigan, USA) to confirm that the value was within the target HR range. The Karvonen formula (1957) [31] was used to calculate HR reserve (HRR, estimated maximal HR − resting HR) and target HR during exercise ((HRR × given percentage of training intensity) + resting HR)). The experimental designs are illustrated in Figure 1.

### 2.3. Kanizsa Triangle Task

Before and following the exercise, participants were prepared for the Kanizsa triangle task. The task was to examine spatial and temporal attention perceived by visual information [32]. We used the computerized Kanizsa triangle task using a built-in software of the EEG analyzer (Telescan, LAXTHA, Daejeon, Korea). Participants were seated in front of the computer screen with a keyboard resting on a desk and were instructed to perform the task. Participants were instructed to press buttons that were relevant to the direction of the target triangle either appearing inside the Kanizsa triangle (valid cue) or contralateral to the Kanizsa triangle (invalid cue) as quickly and accurately as possible. A total of 260 target triangle trials consisted of 140 valid cue, 60 no cue, and 60 invalid cue were presented for each participant. Participants were instructed to press the leftward arrow (←) for a valid cue and press the rightward keyboard (→) for an invalid cue. Participants were also instructed not to respond to a cue without a Kanizsa triangle (no cue). The duration of the test was 10 min. Six alternate versions of the Kanizsa test comprising unique sets of cues were counterbalanced across experimental interventions (i.e., seated-rest, exercise with no color, blue, green, red, and yellow). Participants underwent a familiarization session before starting the test. The number of practice trials was standardized (1 trial each) so that participants performed the same number of practice trials before each condition. The duration of the practice trial was 2:42 min including 65 trials (35 valid cue, 15 no cue, and 15 invalid cue). Both accuracy (% correct) and RT (millisecond; ms) were recorded. The description of the task is illustrated in Figure 2.

### 2.4. Electroencephalography Measurements

EEG activity was recorded with Ag–AgCl electrodes at frontal (Fz) and central (Cz) midline locations referenced to linked electrodes placed on the left and right earlobes, with a forehead ground using 31 channel Poly G-A (LAXTHA, Daejeon, Korea). EEG activity was recorded with electrodes attached with paste and tape at each site. After completion of data collection, EEG signals were analyzed using software (Telescan, LAXTHA, Korea). EEG activities during incorrect trials were excluded from the analysis to isolate activation associated with only the correct response. While exercise could influence error-related processing, the Kanizsa task is relatively easy to perform and produces few error trials and is not designed to effectively address error-related processing.

The data were re-referenced to the average of the left and right mastoids and band-pass filtered with a low-pass frequency of 0.5 Hz and a high-pass frequency of 20 Hz. A 60 Hz notch filter was also applied to remove potential artifacts. ERP component amplitudes were assessed with a computer-assisted peak detecting system. The stimulus-locked epochs acquired for the Kanizsa test were extracted offline from 200 ms pre-stimulus onset to 1500 ms post-stimulus onset, and the period from −100 to 0 ms pre-stimulus onset was used as the baseline. Peak amplitudes and latencies were measured automatically. The mean amplitudes of P2, N2b, and P3a were derived from the mean amplitude values within the corresponding time windows. The P2, N2b, and P3a peak amplitudes and latencies were measured in the different waves of the attended condition. P2 component was defined as the largest positive peak occurring between 120–270 ms at Cz, N2b was defined as the largest negativity occurring between 200–380 ms at Cz, and P3a as the largest positivity between 250–280 ms at Fz and Cz.

### 2.5. Statistical Analysis

Shapiro Wilk test was administered to determine normality. The behavioral (i.e., accuracy and RT) and ERP waveforms (i.e., P2, N2b, P3a amplitude) were analyzed using repeated measures of ANOVA to determine the main effects of the session (pre vs post-exercise) and session (pre vs post-exercise) × intervention (seated-rest, no color, blue–green, red, yellow) interactive effects for all outcome variables. When significant session × intervention interactions were identified, paired sample *t*-tests were performed to detect differences before and after color stimulation. One-way ANOVA was also used to compare the delta values (the difference between before and after color stimulation) of each color stimulation. Bonferroni post-hoc analyses were performed when there was a significant difference. Partial eta squared (η^2^_p_) was used to assess the effect size. The statistical significance was determined using a two-tailed alpha = 0.05. All statistical tests were conducted using SPSS (v. 24.0) (IBM, New York, NY, USA). 

## 3. Results

### 3.1. Behavioral Indices 

There was no significant interaction of session (pre vs. post exercise) × intervention (seated-rest, no color, blue green, red, yellow) and the main effect of session for accuracy rate in both the valid and invalid cue. Results of valid and invalid cue RTs are presented in Figure 3. For the valid cue RT, there was a significant session × intervention interaction (F(5,126) = 9.998, *p* = 0.001, η^2^_p_ = 0.284). There was also a main effect of session (F(1,126) = 196.115, *p* = 0.001, η^2^_p_ = 0.609). As we expected, the pre-post (session) difference in seated-rest RT was not significant (1127 ± 162 ms vs. 1126 ± 162 ms, *p* = 0.937, η^2^_p_ = 0.502). However, RTs after exercise were significantly shorter in no color (1128 ± 190 ms vs. 1025 ± 170 ms, *p* = 0.001, η^2^_p_ = 0.075), blue (1118 ± 150 ms vs. 970 ± 143 ms, *p* = 0.001, η^2^_p_ = 0.203), green (1137 ± 199 ms vs. 948 ± 152 ms, *p* = 0.001, η^2^_p_ = 0.222), red (1125 ± 173 ms vs. 1012 ± 148 ms, *p* = 0.001, η^2^_p_ = 0.110) and yellow (1135 ± 184 ms vs. 1002 ± 149 ms, *p* = 0.001, η^2^_p_ = 0.136). The comparison of pre–post RT differences among interventions were consistently significant. There were significantly shorter RTs in no color (*p* = 0.005), blue (*p* = 0.001), green (*p* = 0.001), red (*p* = 0.002) and yellow (*p* = 0.001) than in seated rest. However, there were no significant differences in RTs between interventions (Figure 3A).

Similarly, there was a significant session × intervention interaction for invalid cue RT (F(5,126) = 12.356, *p* = 0.001, η^2^_p_ = 0.329). Main effect of session on invalid cue RT was also significant (F(1,126) = 210.151, *p* = 0.001, η^2^_p_ = 0.625). As we hypothesized, the pre-post difference in seated-rest RT was not significant (1545 ± 214 ms vs. 1549 ± 194 ms, *p* = 0.687, η^2^_p_ = 0.506). Conversely, significantly shorter invalid cue RTs were observed after exercise with no color (1550 ± 223 ms vs. 1406 ± 212 ms, *p* = 0.001, η^2^_p_ = 0.099), blue (1545 ± 233 ms vs. 1362 ± 209 ms, *p* = 0.001, η^2^_p_ = 0.146), green (1541 ± 212 ms vs. 1284 ± 169 ms, *p* = 0.001, η^2^_p_ = 0.310), red (1569 ± 179 ms vs. 1442 ± 213 ms, *p* = 0.002, η^2^_p_ = 0.095), and yellow (1534 ± 210 ms vs. 1301 ± 225 ms, *p* = 0.001, η^2^_p_ = 0.229). The comparison of pre–post RT differences among interventions were significant. The decrease in RT was significantly greater in no color (*p* = 0.001), blue (*p* = 0.001), green (*p* = 0.001), red (*p* = 0.001) and yellow (*p* = 0.001) than seated rest. In addition, the decrease in RT was significantly greater in green (*p* = 0.013) and yellow (*p* = 0.026) than in red, but there was no significant difference between other interventions (Figure 3B). 

### 3.2. Event-related Potential Data

#### 3.2.1. P2 Amplitude

Average event-related potential waveforms of electrodes for mean P2 amplitudes during the Kanizsa triangle task in seated-rest and pre and post-exercise with five different interventions (i.e., colors) are presented in Figure 4. The analysis of valid cue P2 amplitude revealed a significant session × intervention interaction, significant main effects of the session (Table 2). There were no significant changes in the valid cue P2 amplitude after the seated-rest session. The valid cue P2 amplitude was significantly increased after exercise with no color, blue, green, and yellow (Figure 4). Furthermore, these changes were significantly different among interventions (Table 2). Specifically, the increase in P2 amplitude was significantly greater in green than in seated rest (*p* = 0.025), no color (*p* = 0.008), red (*p* = 0.001), and yellow (*p* = 0.016). However, this increase was significant smaller in red than seated rest (*p* = 0.008), no color (*p* = 0.023), blue (*p* = 0.001), and yellow (*p* = 0.013).

In the invalid cue P2 amplitude analyses, there was also a significant interaction between session and intervention and the main effect of session on invalid cue P2 amplitude (Table 2). There were also no significant changes in the invalid cue P2 amplitude following the seated-rest session. All interventions consistently increased the invalid cue P2 amplitude (Figure 4). These changes were significantly different among interventions (Table 2). The increase in seated rest was significantly less than no color (*p* = 0.012), blue (*p* = 0.017) and green (*p* = 0.001), but the increase was significantly greater in green than in red (*p* = 0.006), and yellow (*p* = 0.004).

#### 3.2.2. N2b Amplitude

Average event-related potential waveforms of electrodes for mean N2b amplitudes during the Kanizsa triangle task in seated-rest and pre and post-exercise with five different interventions are presented in Figure 4. There was a significant session × intervention interaction on the valid cue N2b amplitude (Table 2). The main effect of the session on valid cue N2b amplitude was also significant. No changes in the valid cue N2b amplitude were found in response to seated-rest. While green significantly increased the valid cue N2b amplitude after exercise, other colors did not significantly change (Figure 4). These changes were also significantly different among interventions (Table 2). The increase in N2b amplitude was significantly greater in green than in seated rest (*p* = 0.004), no color (*p* = 0.008), blue (*p* = 0.001), red (*p* = 0.001) and yellow (*p* = 0.046).

While there was a significant interaction effect of session × intervention on invalid cue N2b amplitude (Table 2). The main effect of the session on invalid cue N2b amplitude was not significant. There were no changes in the invalid cue N2b amplitude after seated-rest. Green significantly increased the invalid cue N2b amplitude, whereas other colors did not significantly change (Figure 4). These changes were significantly different among colors (Table 2). The increase was significantly greater in green than no color (*p* = 0.004), blue (*p* = 0.024), red (*p* = 0.001), and yellow (*p* = 0.014).

#### 3.2.3. P3a Amplitude

Average event-related potential waveforms of electrodes for mean P3a amplitudes in Cz region during the Kanizsa triangle task in seated-rest and pre and post-exercise with five different interventions are presented in Figure 4. The results of valid cue P3a amplitude are shown in Table 2. There was a significant interaction between the session and intervention; and a significant main effect of the session on P3a amplitude. There were no changes in the valid cue P3a amplitude in both Fz and Cz regions for the seated rest session. However, blue, green, yellow and no color significantly increased after exercise in both Fz and Cz regions, but red significantly increased in only Cz region (Figure 4). These changes were significantly different among colors (Table 2). In Fz region, the increase in seated rest was significantly less than no color (*p* = 0.013), blue (*p* = 0.001), green (*p* = 0.001) and yellow (*p* = 0.037). The increase was significantly greater in green than no color (*p* = 0.037), red (*p* = 0.001), and yellow (*p* = 0.002). However, the increase in red was significantly less than seated-rest (*p* = 0.001), no color (*p* = 0.011) and blue (*p* = 0.001). In Cz region, the increase in seated-rest was significantly less than no color (*p* = 0.021), blue (*p* = 0.001) and green (*p* = 0.001). The increases in green were significantly greater than red (*p* = 0.001) and yellow (*p* = 0.004). However, the increase in red was significantly less than No color (*p* = 0.016) and blue (*p* = 0.002).

Similarly, the analysis of invalid cue P3a amplitude revealed a significant interaction between the session and intervention, and the main effect of session (Table 2). Invalid cue P3a amplitude was significantly increased in blue, green, no color after exercise in both Fz and Cz region, but red and yellow significantly increased in only Fz region (Figure 4). There were no significant changes after seated-rest. Furthermore, these changes were significantly different among interventions (Table 2). In the Fz region, the increase in seated rest was significantly less than no color (*p* = 0.002), blue (*p* = 0.006), green (*p* = 0.001) and yellow (*p* = 0.034). The increase was significantly greater in green than no color (*p* = 0.021), blue (*p* = 0.025), red (*p* = 0.001) and yellow (*p* = 0.001). In Cz region, the increase in seated-rest was significantly less than no color (*p* = 0.002), blue (*p* = 0.018), green (*p* = 0.001). The increases in green were significantly greater than in no color (*p* = 0.012), blue (*p* = 0.036), red (*p* = 0.001) and yellow (*p* = 0.001); red was significantly greater than seated-rest (*p* = 0.001). However, invalid cue P3a amplitude during red was significantly lower than no color (*p* = 0.026) and blue (*p* = 0.008); yellow was significantly lower than blue (*p* = 0.020).

## 4. Discussion

In the present study, the effects of different acute visual stimulations during a single bout of treadmill exercise on attention processes were examined. We found that acute visual stimulation exercise improved behavioral performance and neural processing during an attention task. The improved visual search attention, attentional capture, and attentional orienting following exercise were most pronounced during exposure to green stimulus compared to the other color stimuli. No color, blue and yellow stimulating exercise have improved visual search attention and attentional orienting more than under red. 

Consistent with previous findings [30,33,34], performing a single session of the exercise was associated with shorter RT compared to rest. Our results further revealed that acute visual stimulation exercise effectively improved attention task RT. Interestingly, these changes were significantly larger in green and yellow than red in invalid cue RT. These results suggest that a single bout of exercise during the presentation of color stimuli yielded improved behavioral attention performance, while green and yellow stimuli have more beneficial effects compared with red stimuli. RT is a reliable indicator to examine the sensory stimulus processing speed and how quickly the central nervous system responds by executing the motor system [35]. Llorens and colleagues [30] suggested that performing a single session of exercise results in shorter RT relative to rest during the spatial attention task. Despite using different cognitive tests from the present study, previous exercise literature using the flanker [36] and the Stroop task [33] demonstrated significantly shorter RT after the completion of acute exercise. This consistency in faster RT following an acute bout of exercise reflects enhanced efficiency in the prefrontal cortex that leads to faster speed decision making and response selection during the cognitive test [37]. Therefore, our finding suggests that green and yellow stimuli during exercise are associated with greater improvements in the speed of decision making and response selection relative to red.

The valid and invalid cue P2 amplitude differed by colors presented during a single session exercise relative to seated-rest. These changes were significantly larger in green than other colors. In addition, valid cue P2 amplitude was larger in blue and yellow relative to red. No color, blue and yellow stimuli have more beneficial effects compared with red stimuli. Our finding aligns well with the evidence reported by Zhou and Qin [38] who demonstrated that acute moderate-intensity aerobic exercise enhanced attentional resources related to perceptual processing through greater P2 amplitude. Research regarding emotions exhibited that positive emotional stimuli have more beneficial effects compared with negative emotional stimuli on the P2 component [39]. Colors have different stimuli for emotions [10], such as that total mood disturbance and perceived exertion were lower during simulated green exercise in the green condition compared to grey and red [40]. Rogerson and Barton [41] showed that general decreases in significant improvements in directed attention were observed in the nature condition (e.g., greenery scene) but did not in the other (e.g., city scene) or control conditions (e.g., without visual stimuli). This suggests exercise with green stimulus promotes quick recovery from directed attention fatigue, thus replenishing an important resource for performing cognitively demanding workplace tasks [41]. Other previous works have found an association between bright red and positive mood [42,43], indicating that the brightness of the color also affects mood. Our results indicate that green color during exercise tends to have additive effects on attention but more studies need to be done to robustly evaluate this hypothesis.

We found that while no color, blue, red, and yellow stimuli exercise had no effects, green stimuli exercise increased the valid and invalid cue N2b amplitude. These results indicate that exercise during the presentation of green stimuli is associated with improved attentional capture. The relationship between exercise and N2 amplitude has not been fully elucidated [38,44,45,46]. In our previous study, the combination of visual stimulation and exercise better regulates the activity of the anterior cingulate cortex (ACC) in the prefrontal cortex than exercise alone, suggesting enhanced selective attention [44]. N2 is strongly implicated in the ACC activity, which is part of the brain network of the prefrontal cortex, and has been suggested as an important indicator of the self-regulation of cognition and emotion [47,48,49]. However, other studies reported that the acute moderate-intensity exercise general decreases in N2 amplitude across scalp sites during flanker task and Stroop task in young adults [27,38]. This suggests that acute exercise might directly limit ACC activity smaller the N2 amplitude [50]. Importantly, it has been suggested that the environment (e.g., outdoor) during exercise promotes attention restoration, independent of physiological influence [41]. Given that green has the role of retirement and relaxation effect [10], relaxation exercise environment may have sensitively stimulated the activity of ACC in the prefrontal cortex of the brain, thereby effectively increasing attention capture than other colors. 

In addition to the P2 and N2b, the valid and invalid cue P3a amplitudes were increased after visual stimulation exercise, with significantly larger after green stimulus than other colors in Fz region. However, the valid cue/invalid amplitude for Cz region was larger in no color and blue than red. These results indicate that different visual stimulation during exercise engenders differential effects on attentional orienting. P3a amplitude is an index of attentional orienting, with increased amplitude reflecting greater focal attention [16]. Pontifex et al. [51] showed that fitness is not related to attentional orienting, as no cardiorespiratory fitness-related changes occurred in the P3a amplitude or latency. The research of Bullock et al. [52] shows that exercise-induced shorten of P3a latent, can increase arousal availability of cognitive resources for suppressing the response to task-irrelevant distractor stimuli. Previous literature elucidated that green stimuli increase P300 amplitude, suggesting that the human physiological system is less stressed and more relaxed when it is exposed to these color stimuli [12,53]. In support, green stimulus leads to improved performance during brain–computer interface spatial attention test [54]. Taken together, our study suggests that green color during exercise may modulate the brain activity that is related to attentional orienting and whether or not this change in brain activity translates into behavioral performance should be robustly tested using a larger sample.

To further our understanding of the physiological mechanisms responsible for the effects observed in the present study, future studies of acute exercise effects on the color vision might include aspects of both ocular and cerebral blood flow. As regards the ocular blood flow, there is evidence reporting the increase in blood flow in the eye due to exercise [55] which could differentially influence the acute exercise effects in different color conditions. As regards the cerebral blood flow, the results regarding the effects of an acute bout of exercise are mixed. Some reported that cerebral blood flow level is maintained during exercise [56], others showed conflicting results regarding the relationship between acute exercise and changes in the cerebral blood flow. For example, studies showed that acute exercise is associated with decreased [57], increased [58], and no changes [59] in the cerebral blood flow. Thus, it remains unclear if blood flow increases in the eye co-occur with increases in blood flow of the visual cortex and other brain regions that are related to spatial attention.

The limitations of our study are the relatively small sample size in the current study. Future studies need to replicate the findings observed in the present study by using a larger sample size. Second, only colors were used as stimuli in the present study. Future studies need to the impacts of other stimuli (e.g., nature view) during exercise on spatial attention. Third, the participants consisted of healthy younger adults with little representation of racial diversity; all participants were Asian. The homogeneous characteristics of the group limit the generalizability of the findings. Next, an important direction for the future is to examine if these effects are replicated in people of different age groups or clinical patients with spatial attentional issues. The present study warrants caution when interpreting the results due to small effect sizes. Despite these small effect sizes, consistency of the magnitude and direction of these effects gives confidence that these findings are not merely spurious. In addition, there were inconsistencies for some color stimulation in the exercise-related changes in RT and ERP amplitudes. It is to consider that ERPs and RTs do not reflect overlapping processes. ERP amplitude reflects a series of cognitive operations that take place at the central level in the brain. Instead, RT is a behavioral response that is the result of processes extending from the central to the peripheral nervous system. Therefore, it is reasonable to posit that changes in ERP amplitudes may not necessarily reflect the changes in RT. Furthermore, the present study is limited to answer the physiological mechanism to explain the differences in brain activation during valid and invalid task conditions. Future studies need to explore what drives the differences in activations in the brain during valid and invalid conditions after exercise with different visual stimulations. 

## 5. Conclusions

In summary, these findings suggest that a single bout of acute visual stimulation exercise may maintain beneficial effects induced by exercise on brain networks related to controlling attention allocation during the performance of visual–spatial task. Additionally, the presentation of colors during physical activity differentially affects attention processes. Particularly, green stimulus carries beneficial effects induced by exercise facilitating greater neural networks and leading enhancement in cortical activity during the process of spatial attention task. Blue, yellow and no color stimuli also carry the beneficial effects induced by exercise facilitating an attentional capture and attentional orienting. Given the paucity of understanding about the link between visual stimuli during exercise and attention processes, the present study provides a novel insight into the influence of acute visual stimulation exercise on human attention processes. Examining changes in spatial attention during exercise with color environment provides evidence for implementing strategies for the optimal environment during exercise and sports. Future research needs to recruit a larger sample of participants with different age groups and utilize different exercise intensities to broaden our scope of understanding regarding the effects of stimuli during exercise on spatial attention.

## Figures and Tables

**Figure 1 ijerph-18-01107-f001:**
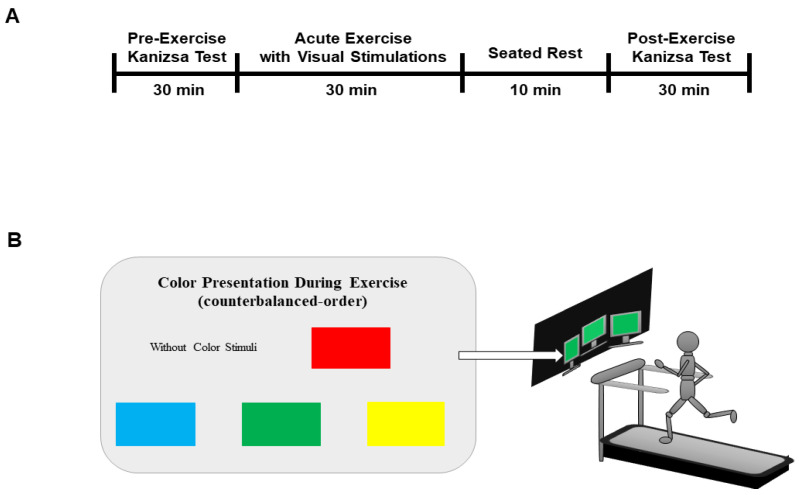
The experimental design of this study: (**A**) After performing the pre-exercise Kanizsa test, participants underwent a 30-min exercise with a color stimulus that was followed by a 10-min seated cool-down. Upon completing the cool-down, participants performed the post-exercise Kanizsa test; (**B**) color stimuli were presented on a monitor screen (in a counterbalanced order across participants), situated 3 m in front and on both sides of the treadmill. To provide the same environment providing the stimuli, the wall behind the monitors was covered with black cloth and the light was off.

**Figure 2 ijerph-18-01107-f002:**
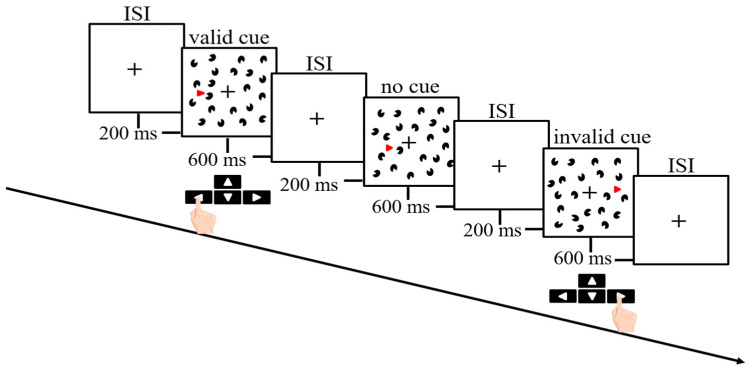
Illustration of the Kanizsa test used in the present study. The target triangle either appeared inside the Kanizsa triangle (valid cue), contralateral to the Kanizsa triangle (invalid cue), or in a cue without the Kanizsa triangle (no cue). The participants were instructed to indicate the direction of the target triangle (right or left). ISI: Interstimulus interval.

**Figure 3 ijerph-18-01107-f003:**
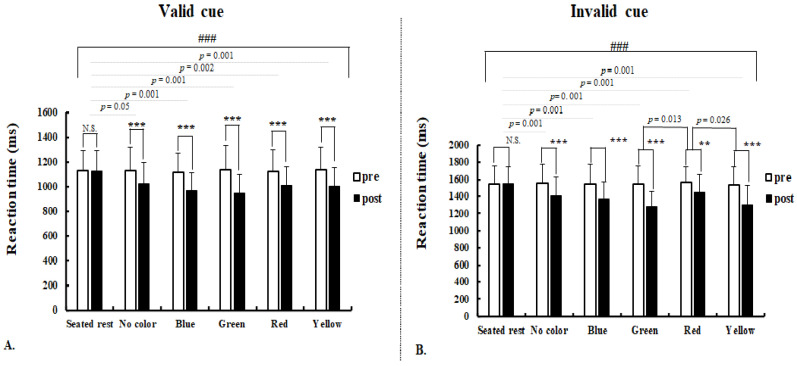
Response time (RT) during the Kanizsa triangle task in seated-rest and pre and post exercise with five different interventions (i.e., colors). (**A**) valid cue; (**B**) invalid cue. *Note*: ***, significant difference in pre vs. post at *p* < 0.001; **, significant difference in pre vs. post at *p* < 0.05; ###, significant interaction on RT changes between five interventions at *p* < 0.001; N.S., non-significant (*p* > 0.05).

**Figure 4 ijerph-18-01107-f004:**
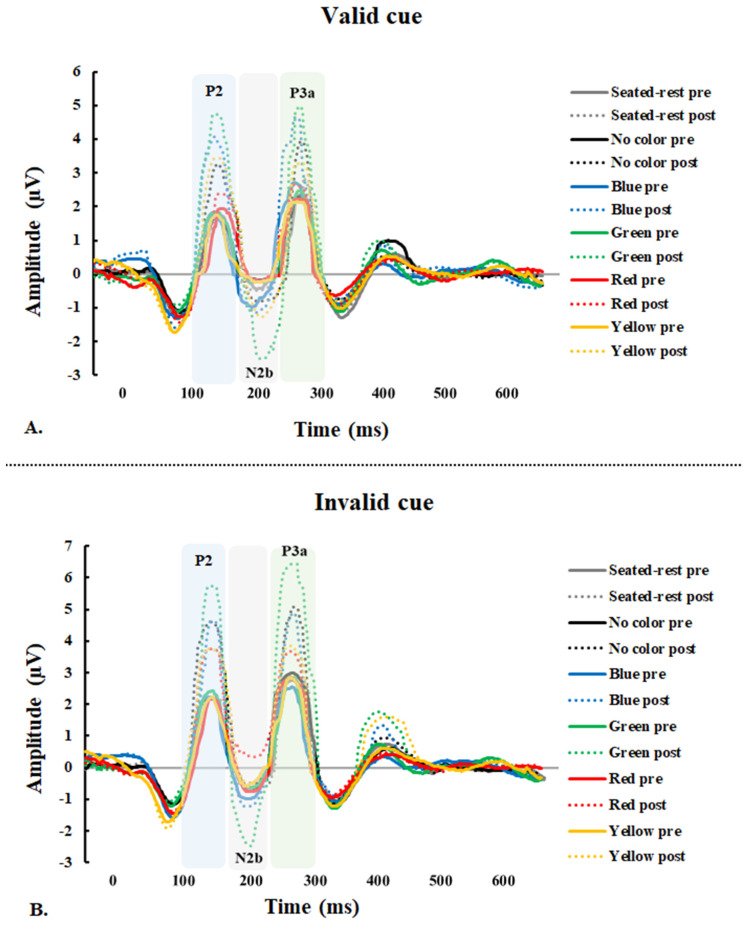
Average event-related potential waveforms of electrodes in Cz for P2, N2b, and P3a amplitudes during the Kanizsa triangle task in seated-rest and pre and post-exercise with five different interventions. (**A**) valid cue; (**B**) invalid cue.

**Table 1 ijerph-18-01107-t001:** Demographic information of study participants.

Variable	Values Are Mean ± Standard Deviation
Sample Size (n)	22 (10 Females)
Age (years)	24.0 ± 3.0
Height (cm)	169.91 ± 7.39
Weight (kg)	61.06 ± 9.42

**Table 2 ijerph-18-01107-t002:** Event-related potential on the Kanizsa Triangle Task for seated-rest and pre and post-exercise in five different interventions.

Valid Cue							Invalid Cue					
Group	Pre	Post	Session *p* (η^2^_p_)	Session × Intervention *p* (η^2^_p_)	Session Effect *p* (η^2^_p_)	Pre-Post Difference Comparison *p* (η^2^_p_)	Pre	Post	Session *p* (η^2^_p_)	Session × Intervention *p* (η^2^_p_)	Session Effect *p* (η^2^_p_)	Pre-Post Difference Comparison *p* (η^2^_p_)
**P2 Amplitude (μV)**												
Seated rest	1.80 ± 1.83	1.75 ± 1.63	0.670 (0.508)	0.001 (0.326)	0.001 (0.507)	0.001 (0.653) G > S, R, N, Y; S, N, B, Y > R	2.21 ± 1.46	2.24 ± 1.62	0.790 (0.505)	0.041 (0.228)	0.001 (0.465)	0.050 (0.454) S < N, B, G G > R, Y
No color	1.67 ± 2.00	3.25 ± 1.72	0.001 (0.725)				2.18 ± 1.92	4.59 ± 1.80	0.001 (0.820)			
Blue	1.71 ± 1.55	4.06 ± 1.53	0.001 (0.860)				2.26 ± 1.73	4.37 ± 1.68	0.001 (0.809)			
Green	1.82 ± 1.78	4.72 ± 1.87	0.001 (0.387)				2.41 ± 1.60	5.72 ± 1.71	0.001 (1.000)			
Red	1.93 ± 1.82	2.37 ± 1.71	0.090 (0.570)				2.14 ± 1.58	3.74 ± 2.74	0.023 (0.694)			
Yellow	1.73 ± 1.85	3.43 ± 1.83	0.001 (0.743)				2.21 ± 1.53	3.75 ± 1.70	0.002 (0.750)			
**N2b Amplitude (μV)**												
Seated rest	−0.45 ± 2.10	−0.49 ± 2.09	0.714 (0.505)	0.001 (0.165)	0.001 (0.125)	0.004 (0.582) S < N, B, G, R, Y; G > N, B, R, Y	−0.70 ± 2.1.9	−0.70 ± 1.99	0.992 (0.500)	0.002 (0.139)	0.492 (0.004)	0.003 (0.621) G > N, B, R, Y; S > R
No color	−0.17 ± 2.47	−0.80 ± 2.32	0.189 (0.574)				−0.74 ± 2.16	−0.49 ± 2.67	0.638 (0.524)			
Blue	−0.98 ± 2.11	−1.13 ± 2.00	0.704 (0.521)				−1.00 ± 2.16	−1.22 ± 2.05	0.624 (0.529)			
Green	−0.23 ± 2.14	−2.50 ± 2.10	0.001 (0.776)				−0.61 ± 2.20	−2.46 ± 3.08	0.001 (0.687)			
Red	−0.21 ± 2.13	−0.17 ± 1.71	0.935 (0.623)				−0.76 ± 2.28	0.34 ± 2.76	0.112 (0.621)			
Yellow	−0.24 ± 2.37	−1.28 ± 2.34	0.074 (0.047)				−0.50 ± 2.19	−0.57 ± 2.69	0.907 (0.508)			
**P3a Amplitude (μV) in Fz Region**												
Seated rest	2.35 ± 0.96	2.27 ± 0.96	0.128 (0.521)	0.001 (0.334)	0.017 (0.548)	0.001 (0.588) S < N, B, G, Y; G > N, B, R, Y; N, B > R N, B > Y B > R, Y	2.81 ± 1.27	2.80 ± 1.41	0.951 (0.502)	0.001 (0.284)	0.001 (0.505)	0.001 (0.653) S < N, B, G, Y; G > N, B, R, Y
No color	2.22 ± 1.31	4.05 ± 1.49	0.001 (0.822)				2.83 ± 1.46	5.20 ± 1.49	0.001 (0.872)			
Blue	2.48 ± 1.05	4.68 ± 1.47	0.001 (0.888)				2.66 ± 1.08	5.08 ± 1.28	0.001 (0.926)			
Green	2.23 ± 1.23	5.00 ± 1.22	0.001 (0.945)				2.94 ± 1.45	6.83 ± 1.45	0.001 (0.971)			
Red	2.11 ± 1.09	2.84 ± 1.41	0.033 (0.659)				2.73 ± 1.51	3.85 ± 2.13	0.047 (0.666)			
Yellow	2.34 ± 0.99	3.66 ± 1.88	0.006 (0.733)				2.73 ± 1.25	4.50 ± 2.62	0.020 (0.729)			
**P3a Amplitude (μV) in Cz Region**												
Seated rest	2.21 ± 1.16	2.42 ± 1.00	0.125 (0.555)	0.001 (0.236)	0.001 (0.448)	0.001 (0.560) S < N, B, G; G > R, Y: N, B > R	2.77 ± 1.08	2.88 ± 1.35	0.205 (0.525)	0.001 (0.299)	0.001 (0.474)	0.001 (0.730) S < N, B, G; G > N, B, R, Y
No color	2.29 ± 1.27	3.91 ± 1.46	0.001 (0.799)				2.97 ± 1.25	5.05 ± 1.35	0.001 (0.871)			
Blue	2.69 ± 1.06	4.64 ± 1.77	0.001 (0.828)				2.53 ± 1.10	4.87 ± 1.23	0.001 (0.922)			
Green	2.46 ± 1.22	4.97 ± 1.31	0.001 (0.920)				2.83 ± 1.47	6.43 ± 1.35	0.001 (0.964)			
Red	2.22 ± 1.27	2.64 ± 1.32	0.257 (0.591)				2.85 ± 1.58	3.68 ± 2.15	0.097 (0.622)			
Yellow	2.13 ± 1.13	3.30 ± 1.76	0.012 (0.712)				2.81 ± 1.28	3.83 ± 2.35	0.105 (0.468)			

Notes: Exercise comparison pre vs. post within group (by paired sample *t*-tests). session × intervention interaction between exercise and color (by repeated measures of ANOVA). S: seated-rest, N: No color, B: Blue, G: Green, R: Red, Y: Yellow.

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
