# Peer review of "Effects of Acute Visual Stimulation Exercise on Attention Processes: An ERP Study"

_ijerph, 2021, doi:10.3390/ijerph18031107_

Round 1
Reviewer 1 Report
I am now positive about publication, as the authors have now included the relevant control condition. I would like to see more details about the control, as explained below (re Fig. 3 and table 3).
A general comment; the authors have discovered what, to me as a researcher in both visual attention and color vision, is a novel effect. Had they simply been replicating an established result, the no-exercise control would probably not have been necessary - but as their result is both new and unexpected, I think it is essential to include it. I understand that this is a ‘judgement call’ but I tink the paper is greatly strengthened by including it.
Remaining issues.
- Line 175
“For the valid cue RT, there was no significant interaction of session × intervention [F(5,126)=9.998, p=0.001, η2p=0.284]. However, there was a main effect of session [F(1,126)=196.115, p=0.001, η2p=0.609]. “
‘no significant interaction’ should be ‘a significant interaction’ as the critical value for F(5,126) at 0.001 is Fcrit=3.3
- Line 177 grammar (‘were’ missing)
“Further analyses revealed that the valid cue RTs after seated rest (1127±162 ms vs. 1126±162 ms, p=0.937, η2p=0.502), significantly shorter valid cue RTs were found after exercise for all colors as well as no color…”
Suggested fix:
“Further analyses revealed that the valid cue RTs after seated rest (1127±162 ms vs. 1126±162 ms, p=0.937, η2p=0.502) were significantly shorter than valid cue RTs found after exercise for all colors ….
- Line 182,
The authors refer to Fig 3a. However, this figure shows the before-after difference for the exercise group (good), but not for the seated rest group. Figs. 3a and 3b reveal the valid/invalid cue effect only.
BUT this is the critical point; I need to see that the pre-post difference is smaller in the seated group than in the exercise group, either for all of the colors or for just some of the colors, which itself would be interesting.
A plot like Fog 3a and 3b for the control group would be convincing. At the moment the relevant information is rather obscured by presenting it just as F-tests in the text - and given point 1 above, that there is a significant interaction, it is best to be able to see it.
- Table 3 does reveal a comparison for the ERPs, between seated and exercise groups, taken as a whole.
However the break-down by color is again not provided. So again, I cannot tell if exercise modulates the response to some colors more than to other colors.
****
Why might the color matter? Researchers in color vision have found much larger effects in visual cortex of changing illumination on the yellow-blue (daylight locus) axis than on the red-green axis. However, blood flow increases in the eye, due to exercise, might well create larger effect in the red direction. So the seated versus exercise comparison may help to reveal some of the underlying physiology, if the data are provided separately for each color, rather than being lumped together.
Reviewer 2 Report
I would like to commend the authors first, for your creativity in the research design and second, for your thoughtful revisions. This manuscript should open the door for how we conduct rigorous research in more meaningful ways. Non-impaired humans have come to rely on visual cues to make decisions and navigate their environment. Understanding how this information is processed before, during, and after exercise has merit. The findings from this study have clinical and real-world applications.
Reviewer 3 Report
I have a minor observation. After they provide this information (it's discretional, but I think the manuscript would benefit from that), it can be accepted without sending it back for review.
Comment 1 Response: We agree with the reviewer that our hypothesis was not clear and lacked rationale. We have included a clear hypothesis that is centered on green color and rationale at the end of introduction (lines 72-74)
- Observation: Yes, it's clearer now, but you still don't indicate what do you mean with "greater improvement in spatial attention performance and EEG-related responses compared to other colors and no color". Which variable(s) were you expecting to be "improved" (faster Reaction Times, higher Accuracy?) when subjects watched green compared to other colors or no colors, and in which conditions (congruent, incongruent, neutral)? And which are the EEG components that you expect to reflect the "improvement" in the "green" situation? P2, N2b, and P3a Amplitude/Latency, or only one of them (for example P3a for congruent and N2b for incongruent conditions...), basing on existing literature (which you discuss in the Discussion section, lines 232-330?).
Author Response
Please see the attachment.

This manuscript is a resubmission of an earlier submission. The following is a list of the peer review reports and author responses from that submission.
Round 1
Reviewer 1 Report
I am concerned that the design of this research is weak. The authors compare measurements made before and after vigorous exercise, but have no control condition in which the vigorous exercise is replaced by quiet sitting for the same period. Therefore we do not know whether the effect of color on the tasks have anything to do with exercise, or are just an effect of repetition of the test, independent of excercise.
I suggest taking a fresh batch of 20 students and doing exactly the same experiment again, but this time, with quiet sitting instead of exercise. A comparison of color effects on the tasks across the two groups may support the authors' conclusions, and if so, the work will be publishable.
Otherwise the paper is well done and the results are potentially interesting. With a control condition in place, I would be happy to review the paper again for details.
Reviewer 2 Report
The authors of this paper examined the effects of presenting visual stimuli during treadmill running exercise. The testing protocol and data represent novel, preliminary data in this area. I appreciate the authors' conservative assertation that this question has value to determine the optimal training conditions and that this is a first step in the process of that identification, given the lack of a significant interaction effect under most conditions. I thank the authors for the inclusion of the effect size calculation.
There are three distinct limitations to this study: (a) the authors did not account for fitness level, which in this population could influence the behavioral accuracy response, (b) lack of a direct measure of physical activity intensity, and (c) lack of screening for color-blindness.
I would like to hear from the authors if they do have this information and would be willing to share it. First, not everyone exercises at the same exact intensity, given fitness differences. Second, aerobic fitness should have been a covariate in the analysis. I am willing to hear the authors' rationale for what these data were not included, but at the very least these are limitations of the study.
Why train during exercise when someone could play the video game Tetris and improve spatial attention and visuospatial accuracy and RT? Are there added benefits to training in this way, over just playing a video game?
Did you consider analyzing the error rate and the latency post-error? This would be an indicator of inhibitory control, which is slightly beyond the focus of this particular paper, but would add value to your findings.
Did the researchers' screen for color blindness? Red, green and blue light are not detected in individuals who are color-blind.
Reviewer 3 Report
The present study aimed to show that presenting persons with some colors during physical exercise has an impact on cognitive performance, particularly attentional processes. The behavioral results show that exercise improved RTs of an attentional task, while the electrophisiological ones show that P2, N2 and P3a components have an increased amplitude during the task (after performing the physical exercise).
I have the following commentaries and suggestions:
- Introduction: authors hypothesized that different color stimuli during exercise would have different effects on spatial attention and the differential EEG responses. Should be more precise and add the kind of these "different effects" are expected according to behavioral and electrophysiological variables, such as (for example) which color affects which EEG component, and why and how?
- Methodology:
- lines 78-79 "All participants were cognitively asymptomatic, without taking neurological medication or reporting neurological problems". I think you should just say that the participants were healthy subjects, that sentence sounds a bit strange ("cognitively asymptomatic").
- Line 81: substitute "to refrain from eating for four hours" with "to avoid eating for four hours".
- I think that 22 subjects is quite a small N though.
- Lines 113-115: "Participants were instructed to press buttons that were relevant to the direction of the target triangle either appeared inside the Kanizsa triangle (valid cue) 114 or contralateral to the Kanizsa triangle (invalid cue) as quickly and accurately as possible". Change with "...either appearing inside...".
- Lines 141-144: Please provide some details about the time-window baseline and how did you consider the analyzed epochs. Did you perform any automatic or manual ICA analysis during pre-processing or so?
- Line 146: "Shaprio-Wlik test" is written wrong, correct with "Shapiro Wilk"
- Lines 145-154: in section "3.5. Statistical Analysis", you explain that you performed repeated measures ANOVA on the RTs, accuracy and each of the 3 components of ERPs. But since you had many blocks and 5 experimental sessions, I assume that you first calculated the means (or the mode?) for each subject collapsed for all the sessions...is it so? Please, specify it in the text to be sure.
- Another issue is that you say that you determined the main effects of "exercise" and "exercise × color" interactive effects. But you only have 1 condition of exercise: so, what does this variable stand for? Which effect was supposed to express? If all the subjects performed the same exercise, there was only 1 level for this variable. Maybe you evaluated the "session" variable, comparing the mean values of session 1 to the ones of session 2? If so, you should change the name of the variable to "session" instead of "exercise", because it's confusing.
- Line 162: "as well as on color", change to "no color" - Results: generally speaking, I don't see the necessity to explain all the "between colors differences" in the results, since these differences don't explain much of the improvements. I would stick to the fact that green color is the only one which seems to have a significant effect, while red one is the one who shows less difference between session 1 and session 2. For example, what does imply that the difference of pre-post exercise RTs of red with respect to green and with respect to yellow is significant? Does it have an important implication in confirming the hypotheses or as an outcome?
Another issue is that you should beware of the results with quite low eta-squared values (all that is lower than 0,20 is not really powerful). - Discussion: authors say (and conclude) that the vision of a color (expecially green) + the exercise explain your findings that participants performed better in the attentional task after the exercise session. I think that your results show clearly that there is no significant interaction between the color and the exercise for the RTs in the valid cue condition. In fact, you only found a significant main effect of "exercise", which only explain, in my opinion, that the exercise improves the allocation of attention (independently from the presentation of anything while participants were running).
- Line 239: "...orienting than under red" add "more than under red".
- Line 296: "...research shows that demonstrate exercise-induced..."
- Line 297: "...can arousal increase availability..." change with "can increase arousal availability".
- Lines 301-303: "Taken together, our study suggests that green, blue, no color and yellow stimuli may elicit reduced physiological stress during exercise and contributes to maintaining the salutary effects induced by exercise on attentional orienting". No color is actually not a color, so cannot elicit anything because subjects were seeing nothing while doing exercise, so their improvement was due to the exercise and not to the presentation of "no color". It is one of the reasons why I don't agree with your statement that "exercise while visually stimulated with a color improves attentional processes". This is not true, maybe green has some kind of effect, but you still have to perform more studies to evaluate this, but the other colors (and for sure "no color"!) cannot be discussed on the same level.
- Line 304: I suggest changing the following sentence "Despite the strength of our study, this study is not without limitations" with a simpler "The limitations of our study are...".
- Lines 310-311: the following sentence is the same as in lines 305-306, so delete one of them: "Future studies need to investigate the impacts of 310 other stimuli (e.g. nature view) during exercise on spatial attention".
- Line 311-312: "Lastly, the present study 311 recruited relatively small number of participants". I would put this in the beginning instead that as a last thing: it's one of the major limitations. - Conclusions:
- Lines 324-326: you conclude that "Color is indispensable in the natural environment, which means that exercise in the natural environment more than a single exercise improves people's attention processes". I think that your results don't allow to conclude that. It might be, but this is an hypothesis, so you cannot say this as if it was an outcome of your study.
Round 2
Reviewer 1 Report
In experiments with the chronology M1, experimental task, M2, where M1 and M2 are measurements made on the same person, it is usual to include a control of the form M1, control task, M2, in order to assess whether any differences between M1 and M2 are due to the experimental task (here, exercise) or occur because of repeated testing or the elapse of time or the time of day etc., etc.
It seems clear to me that to qualify as a contribution to the science of Exercise, it is essential that a suitable control be used. It is easy to imagine that the various effects of color and so on would happen equally in the exercise group as in a control group involving gentle activity.
The authors need to document that this does not happen, in order to prove that exercise is indeed responsible for the effects that they discovered. If they did this, I would be very enthusiastic for publication, as the outcome of their researchers is otherwise very interesting.